# Explore the possible influence of Sjogren's syndrome on thyroid cancer: A literature data mining and meta-analysis

Fanyong Kong[1], Boxuan Han[2,3], Zhen Wu[4], Jiaming Chen[2,3], Xixi Shen[2,3], Qian Shi[2,3], Lizhen Hou[2,3], Jugao Fang[2,3☉]*, Meng Lian[2,3☉]*

**1** Department of Otorhinolaryngology, Beijing Shunyi District Hospital, Shunyi Teaching Hospital of Capital Medical University, Beijing, China, **2** Department of Otorhinolaryngology Head and Neck Surgery, Beijing Tongren Hospital, Capital Medical University, Beijing, China, **3** Key Laboratory of Otorhinolaryngology Head and Neck Surgery (Capital Medical University), Ministry of Education, China, **4** Department of Thyroid and Breast Surgery, Liaocheng People's Hospital, Liaocheng, Shandong Province, China

☉ These authors contributed equally to this work.

\* fangjugao2@ccmu.edu.cn (JF); lianmeng19861222@163.com (ML)

## Abstract

### Objectives

To explore the potential influence of Sjogren's syndrome (SS) on thyroid cancer (TC).

### Methods

First, a literature data mining (LDM) approach was used to reconstruct functional pathways connecting SS and TC. A meta-analysis was then performed to examine the expression changes of genes mediated by SS using 16 TC case/control expression datasets, with results validated through the TCGA/GTEx dataset. Finally, gene set enrichment analysis (GSEA) and survival analysis using GEPIA2 were conducted on the significant genes.

### Results

Our findings indicate that SS may increase the risk of TC by activating 14 TC promoters (PDCD1, NTRK1, LGALS3, CD274, FOXP3, BCL2, CYP1A1, HMGB1, TGFB1, CCL2, PLA2G7, TFF3, LCN2, and CLDN1) and suppressing three TC inhibitors (MIR145, MIR30C1, and EP300). Four molecules (PLA2G7, TFF3, LCN2, and CLDN1) exhibited significant expression changes in TC patients (LFC > 1 or < -1; p < 2.07E-04), which were confirmed in TCGA/GTEx expression analysis. These results highlight three possible mechanisms—the SS-PLA2G7-CCL2-TC pathway, the SS-LCN2-LGALS3-TC pathway, and the SS-CLDN1-BCL2-TC pathway—that may explain how SS contributes to TC development. Enrichment analysis suggests that SS may affect TC prognosis by regulating leukocytes and tolerance induction. Survival analysis indicates that SS may enhance TC survival through the regulation of the CLDN1 and EGF pathways.

**Data Availability Statement:** All data generated or analyzed during this study are included in this article. The GEO datasets used in this study are available from the GEO database (https://www.

ncbi.nlm.nih.gov/geo/) the corresponding GEO IDs: GSE35570, GSE58545, GSE58689, GSE60542, GSE65144, GSE39156, GSE53157, GSE29265, GSE33630, GSE27155, GSE5364, GSE6339, GSE9115, GSE3678, GSE6004, GSE3467. Further inquiries can be directed to the corresponding author.

**Funding:** This study was financially supported by the Beijing Municipal Administration of Hospitals' Youth Programme in the form of a grant (QMS20210203) received by ML. This study was also financially supported by the Scientific Research Common Program of the Beijing Municipal Commission of Education in the form of a grant (KM202210025014) received by ML. This study was also financially supported by the Beijing Natural Science Foundation in the form of a grant (7252194) received by ML. The funders had no role in the study design, data collection and analysis, decision to publish, or preparation of the manuscript.

**Competing interests:** The authors have declared that no competing interests exist.

## Conclusion

LDM-based pathway analysis highlighted three genetic pathways through which SS may adversely affect TC progression, while SS may enhance TC survival via the CLDN1 and EGF pathways, highlighting the need for further research.

## Introduction

Sjögren's syndrome (SS) is a chronic autoimmune disease that causes dryness of the mouth and eyes, as well as other symptoms such as dry skin, vaginal dryness, chronic cough, numbness in the arms and legs, fatigue, muscle and joint pain, and thyroid problems [1]. SS can also seriously affect other organ systems, such as the lungs, kidneys, and nervous system [2]. Furthermore, it increases the risk of lymphoma by 15% [3]. SS is estimated to affect 0.1% to 4% of the general population, with a higher prevalence in middle-aged women (female-to-male ratio of 9:1) [1, 4, 5].

Thyroid cancer (TC) is a cancer that originates in the cells of the thyroid gland, a gland located in the front of the neck responsible for producing hormones that regulate the body's metabolism [6]. Although TC is one of the least lethal types of cancer, with over 56,000 new cases reported in the United States in 2017 and an estimated 2,010 deaths from thyroid cancer were reported in the same year [7]. The incidence has been increasing since the 1990s, rising from approximately 5.0 to 15.0 per 100,000 in 2014, with a higher incidence rate in women (22.2 new cases per 100,000) [7].

Although the exact cause of TC is unknown, certain risk factors have been identified, such as a family history of TC, exposure to radiation, and specific genetic syndromes [8]. Several studies have suggested a potential association between SS and TC. For instance, one retrospective study reported a significantly higher incidence of TC in patients with SS than in the general population (pooled Standardized Infection Ratio (SIR) = 2.05, 95%CI 1.20–3.48) [9], and several recent studies have also confirmed the increased risk of TC among patients with primary SS [10–12]. However, a definitive causal relationship between SS and TC has yet to be established, and the nature of the SS-TC relationship remains unclear and requires further investigation.

In this study, we utilized extensive literature data mining to construct a molecular pathway driven by SS that potentially affects the development and progression of TC. In addition, we conducted a meta-analysis of RNA expression to examine the levels of SS-driven molecules in patients with TC. We also performed a pathway enrichment analysis to investigate the functions of the molecules linking SS and TC. Our findings provide novel insights into the role of SS as a risk factor for TC.

## Materials and methods

### SS-driven molecules influencing TC

To investigate the potential influence of SS on TC, we performed literature data mining (LDM) to identify SS-driven molecules that also act as upstream regulators of TC. The LDM was carried out using Pathway Studio (version 12.3), which utilizes the natural language processing (NLP) tool MedScan [13]. This process involved mining data from 24 million PubMed abstracts and 3.5 million Elsevier and third-party full-text papers. Each relationship or edge was established based on facts extracted from the literature using NLP technology, supported

by at least one reference. Key information extracted included relation type, relation polarity, reference title, PMID/DOI, and the key sentences where the relationship was identified. First, the downstream targets of SS with polarity (activated or inhibited in SS) were identified. Then TC upstream regulators with polarity (advancing or inhibiting the disease) were also identified. The overlapped entities were used to construct the molecular network connection between SS and TC. The relationship data within the molecular network was extracted from the Elsevier Knowledge graph database (www.pathwaystudio.com), which covers the entire PubMed database, Elsevier publications, and third-party literature, updated as of February 2024. A manual check of the underlying references for each relationship was conducted for quality control.

## TC RNA expression data acquisition

To explore the quantitative changes of SS-driven proteins in the case of TC, we acquired TC RNA array-expression datasets from GEO (https://www.ncbi.nlm.nih.gov/geo/). The term 'Thyroid cancer' was used as the keyword for the search, with 91 series datasets identified. Then, we applied the following criteria to fulfill the purpose of this study, including 1) The data type was RNA expression by array; 2) The organism of the dataset was Homo sapiens; 3) the dataset was with an SS vs. healthy control study design; 4) The sample size≥10. There were 16 datasets that satisfied the selection criteria and were downloadable for expression analysis, as shown in Table 1. The RNA array-expression datasets used in the study were downloaded from GEO (https://www.ncbi.nlm.nih.gov/geo/). It is worth noting that we focused on RNA expression data obtained through arrays rather than NGS data due to compatibility issues with the meta-analysis tool used in this study. Additionally, much of the NGS data available in GEO is incomplete, and the accessible data formats are largely incompatible with our analysis framework. To overcome this limitation, we conducted further analysis of the SS-driven genes that influence TC using TCGA/GTEx thyroid carcinoma (THCA) data.

## Meta-analysis models

A meta-analysis was conducted to assess the impact of TC on SS downstream target genes. Gene expression log2 fold-change (LFC) was estimated using 16 TC expression datasets

**Table 1. The 16 thyroid cancer RNA expression datasets.**

| Dataset GEOID | #Control/#Case | Study Region | Study Age | Sample Organism |
|---|---|---|---|---|
| GSE35570 | 51/65 | Poland | 4 | Homo sapiens |
| GSE58545 | 18/27 | Poland | 4 | Homo sapiens |
| GSE58689 | 18/27 | Poland | 4 | Homo sapiens |
| GSE60542 | 34/33 | Belgium | 4 | Homo sapiens |
| GSE65144 | 13/12 | USA | 4 | Homo sapiens |
| GSE39156 | 16/48 | Belgium | 6 | Homo sapiens |
| GSE53157 | 3/24 | Portugal | 6 | Homo sapiens |
| GSE29265 | 20/29 | Belgium | 7 | Homo sapiens |
| GSE33630 | 45/60 | Belgium | 7 | Homo sapiens |
| GSE27155 | 4/95 | USA | 8 | Homo sapiens |
| GSE5364 | 58/270 | Singapore | 11 | Homo sapiens |
| GSE6339 | 135/48 | France | 12 | Homo sapiens |
| GSE9115 | 4/15 | USA | 12 | Homo sapiens |
| GSE3678 | 7/7 | USA | 13 | Homo sapiens |
| GSE6004 | 4/14 | USA | 13 | Homo sapiens |
| GSE3467 | 9/9 | USA | 14 | Homo sapiens |

presented in Table 1. The LFC for each gene extracted from each dataset was used as input for the meta-analysis, which employed both random-effects and fixed-effect models [14]. The results were compared, and between- and within-study variances were calculated to determine the heterogeneity of the datasets. If the total variance Q was not greater than the expected value of the between-study variances (df), the ISq was set to zero, and the fixed-effect model was selected instead of the random-effects model. The analyses were conducted using Matlab (R2017a version). Significant genes were identified as those with abs (LFC) greater than or equal to 1 and a p-value less than 0.05.

### Analysis of influential factors

A multiple linear regression (MLR) analysis was conducted to determine the potential impact of various factors (such as study date, country of origin, and sample size) on gene expression in TC patients. The expression LFC and the value of each potential influencing factor were used as input for the MLR analysis. The data publication age (current year minus the year of data publication) was used to test the study date, while the index number of each country was used as input to test for country of origin. The total number of samples (number of cases plus number of controls) was used as the sample size. P-values and 95% confidence intervals (CI) were reported for each factor.

### The gene set enrichment analysis for selected targets

To identify potential pathways and functional groups affected by SS and influencing the pathology of TC, gene set enrichment analysis (GSEA) was performed on the significant genes identified in the expression meta-analysis as well as the genes identified from the literature-based analysis described above. This analysis aimed to identify potential functional connections between SS and TC.

### Analysis of SS-driven genes in thyroid carcinoma using TCGA/GTEx data

GEPIA2 is an enhanced web server for large-scale expression profiling and interactive analysis [15]. We performed further analysis of the identified SS-driven genes to investigate their contribution to the pathological development and progression of thyroid carcinoma (TC) using the TCGA/GTEx thyroid carcinoma (THCA) dataset. This dataset includes 512 tumor samples, 59 normal samples from the cancer dataset, and 278 normal samples from the GTEx dataset for comparison. In addition to the meta-analysis using the GEO expression dataset, we further analyzed the expression levels of genes involved in the SS-TC pathway and compared them with the results from the meta-analysis.

Furthermore, the influence of these genes on THCA survival was tested using Survival Analysis of GEPIA2 (http://gepia2.cancer-pku.cn/#survival) and compared with their roles in the pathway, using the log-transformed hazard ratio (HR) to represent their impact on thyroid carcinoma.

## Results

### SS-driven molecules pathway influencing TC

Fig 1 illustrates the network relationships between SS and TC at the molecular level. SS is associated with the upregulation of ten genes/proteins that promote TC, including PDCD1, NTRK1, LGALS3, CD274, FOXP3, BCL2, CYP1A1, HMGB1, TGFB1, and CCL2. Additionally, SS negatively regulates three genes/proteins that inhibit TC, namely MIR145, EP300, and FOXP3. This suggests that SS may adversely affect TC progression and development through

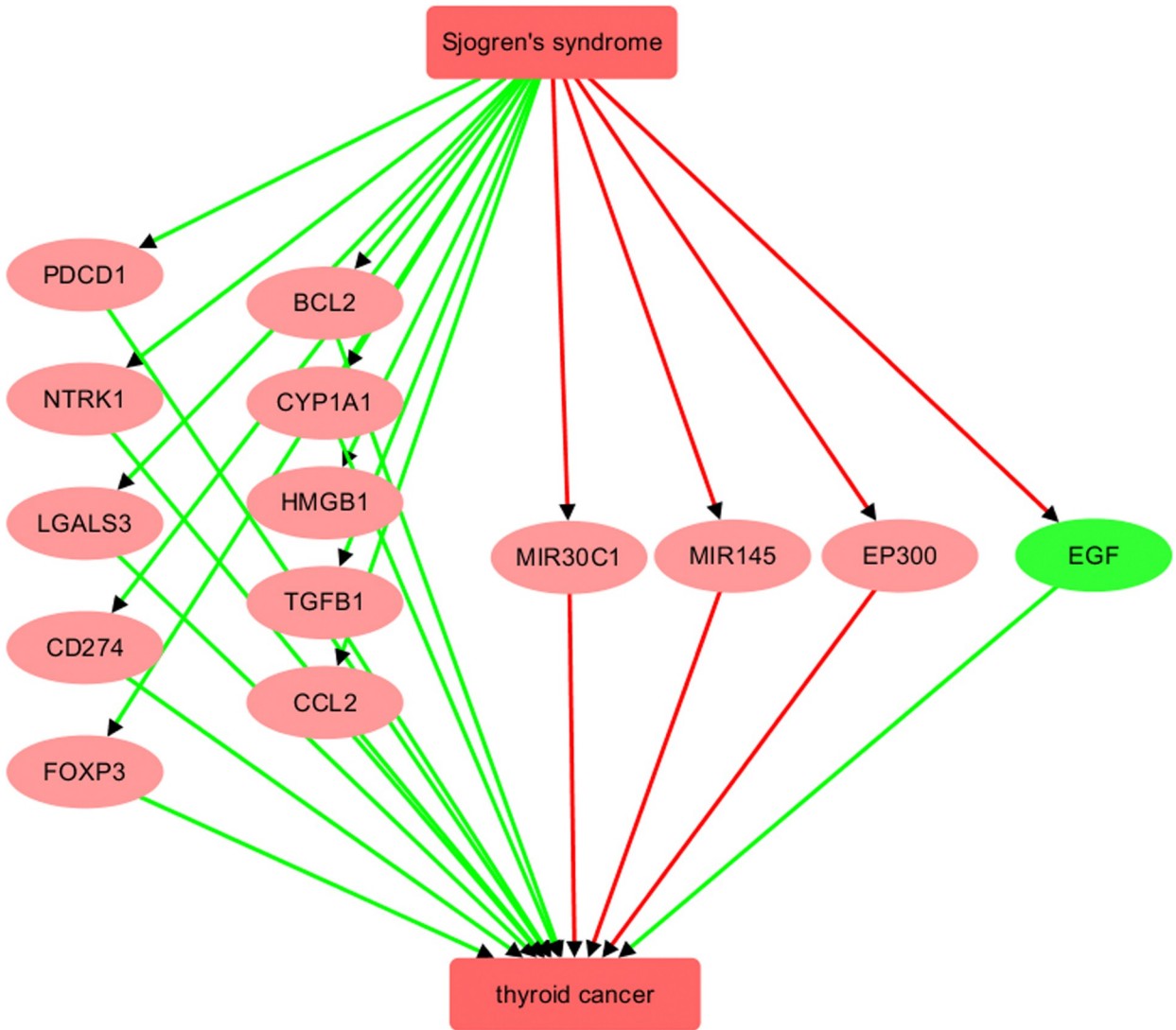

**Fig 1. The Sjogren's syndrome-driven, literature-based molecular pathway influencing thyroid cancer.** Edges in red represent a negative relationship, while those in green represent a positive relationship. A gene in red indicates that SS exerts a detrimental effect on TC through the gene, whereas a gene in green indicates that SS exerts a beneficial effect on TC through the gene.

these 13 genetic pathways. For details on the relationships/edges presented in Fig 1, please refer to S1 Table, which includes information on the supporting references, such as the reference title, PMID, DOI, and the relevant sentences where the relationships were identified. Each relationship is supported by one or more references, with each reference corresponding to a single row in S1 Table.

However, it is important to note that the genes in the SS-TC pathway depicted in Fig 1 did not exhibit significant expression changes in either the meta-analysis of GEO TC expression data or the THCA dataset, as shown in Fig 2A. Furthermore, none of these 13 genes demonstrated a significant impact on the overall survival of thyroid carcinoma ($p(HR) > 0.05$), as presented in Fig 2B. These findings highlight the need for further validation of these genetic pathways.

In addition, Fig 1 shows that SS inhibits a promoter of TC, specifically EGF. Survival analysis revealed that EGF is significantly positively associated with the risk of thyroid carcinoma (p

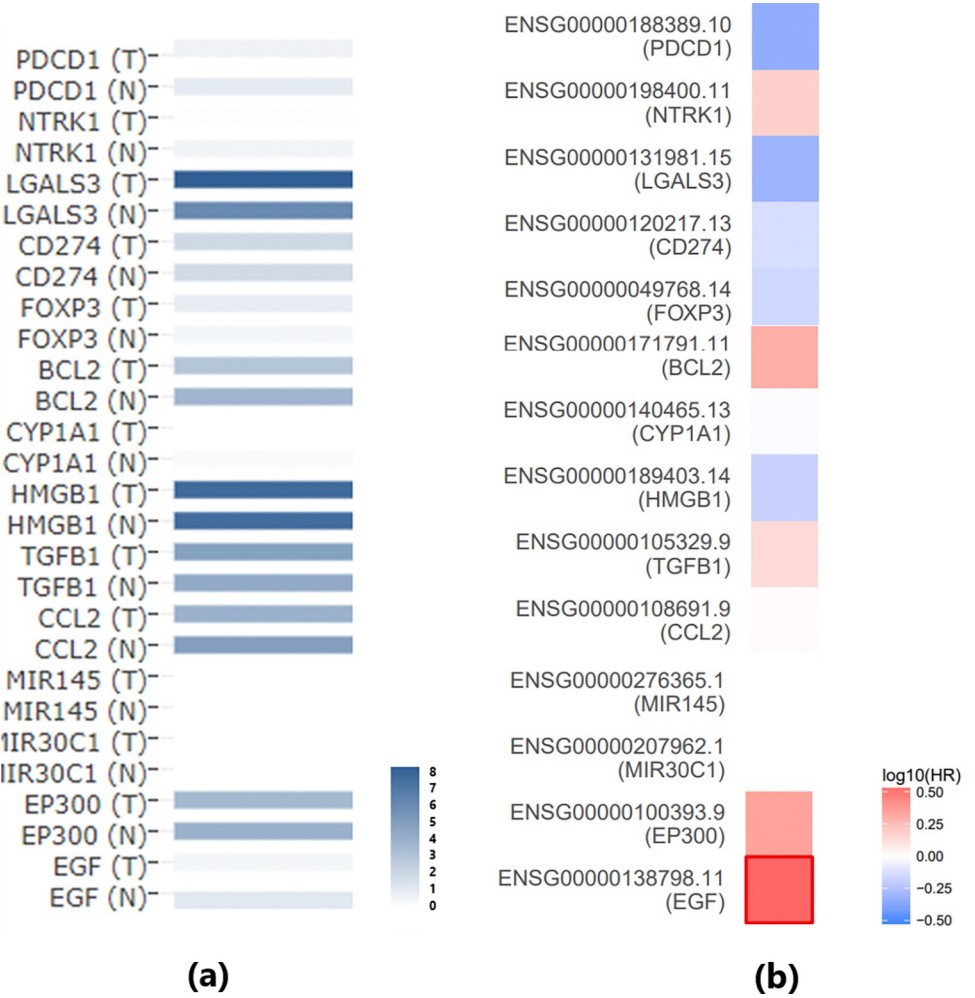

**Fig 2. Expression and survival analysis of the 13 Sjogren's syndrome-driven genes using THCA dataset.** (a) Gene expression of the involved genes using the THCA dataset; (b) Survival map of these genes in thyroid carcinoma. Red indicates that higher expression levels are associated with increased risk (unfavorable prognosis), while blue indicates that higher expression levels are associated with reduced risk (favorable prognosis).

(HR) = 0.03) (Fig 2B). This finding aligns with the SS-EGF-TC pathway presented in Fig 1, suggesting that SS may also play a beneficial role in TC survival, highlighting the complex relationship between SS and TC that warrants further investigation.

## Meta-analysis results

For the 273 SS-driven molecules, 245 were identified in at least one of the 16 TC expression datasets. Except for CD68 and NEAT1, all other genes were identified in at least 10 TC expression datasets. Detailed meta-analysis results for each of these genes are provided in S2 Table. The table includes the meta-analysis model used, the number of datasets (studies) involved, log fold change (LFC; column 'mT'), p-value (column 'p1'), and the results of multiple linear regression (MLR) analysis. Specifically, the columns 'Country', 'nSample', and 'studyAge' provide the p-values from the MLR analysis, indicating the significance of the influence of country of origin, sample size, and study date on a specific gene. Fig 3 illustrates the most significant results of a meta-analysis of the selected TC gene expression studies. A random effects model was used to combine the effect sizes from all individual studies.

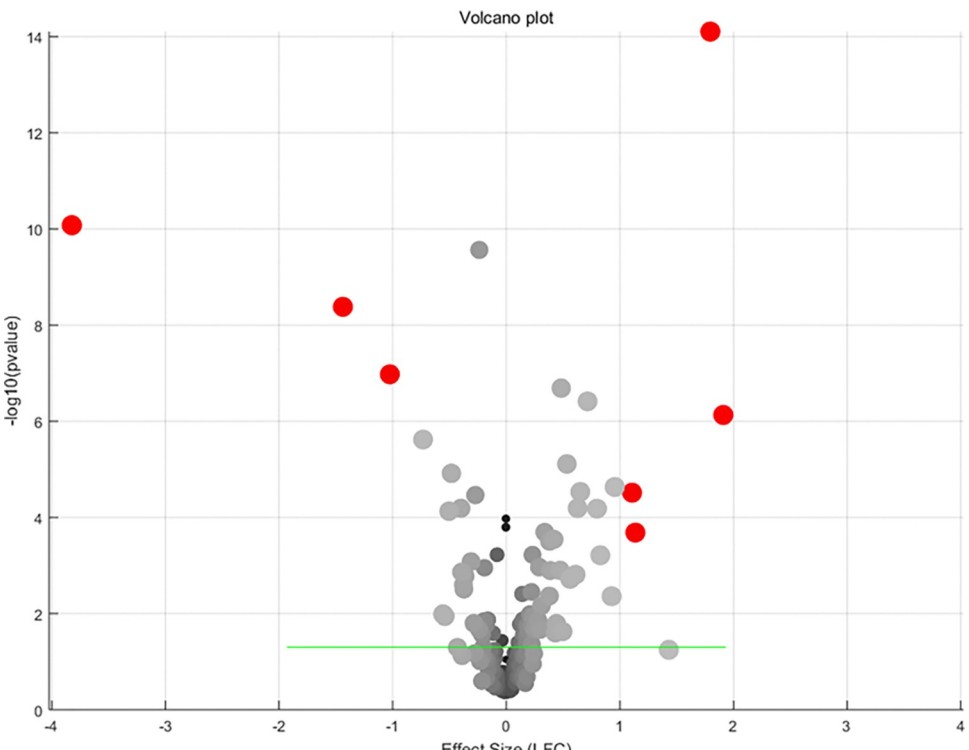

**Fig 3. Volcano plot of the meta-analysis results of 274 Sjogren's syndrome-driven molecules using 16 thyroid cancer RNA expression datasets.**

Table 2 lists all the individual genes that passed the significance filters. Information provided by the table includes whether or not a random or fixed effect model was used in the meta-analysis for that gene, as well as the effect size and p-value. Also included are the number of studies used per gene and the results of multiple linear regression analysis (MLR) performed for various possible confounding factors (country of origin, sample size, and age of study) for those genes for which the random effects model was applied.

For CLDN1 and TIMP1, both genes have a positive effect size, indicating that the expression of these genes is increased in the condition being studied. The p-values for these genes are also very small (7.40E-07 and 7.88E-15, respectively), indicating strong evidence of an association between the gene expression and the condition. For LCN2 and LGALS1, both genes have a positive effect size, but the p-values are less significant than for CLDN1 and TIMP1. This suggests that the evidence for an association between gene expression and the condition is weaker. PLA2G7 and LTF have negative effect sizes, indicating that the expression of these

**Table 2. Genes present a significant change in thyroid cancer patients.**

| Gene Name | Use Random Effects Model | #Study | Effect size | p-value | Country | #Sample | Study age |
|---|---|---|---|---|---|---|---|
| CLDN1 | 1 | 15 | 1.91 | 7.40E-07 | 1.13E-05 | 0.42 | 0.071 |
| TIMP1 | 1 | 16 | 1.80 | 7.88E-15 | 4.50E-03 | 0.57 | 0.44 |
| LCN2 | 0 | 16 | 1.14 | 2.07E-04 | n/a | n/a | n/a |
| LGALS1 | 1 | 15 | 1.11 | 3.07E-05 | 0.17 | 0.38 | 0.79 |
| PLA2G7 | 0 | 15 | -1.00 | 1.06E-07 | n/a | n/a | n/a |
| LTF | 0 | 15 | -1.40 | 4.17E-09 | n/a | n/a | n/a |
| TFF3 | 1 | 15 | -3.80 | 8.38E-11 | 5.51E-03 | 0.08 | 0.062 |

genes is decreased in the condition being studied. The p-values for these genes are very small, indicating strong evidence of an association between the gene expression and the condition. TFF3 has a very large negative effect size, indicating a strong decrease in gene expression in the condition being studied. The p-value is also very small, indicating strong evidence of an association between the gene expression and the condition.

Overall, the results of the meta-analysis suggest that there are multiple genes whose expression is associated with the condition being studied, with varying levels of confidence. However, further studies will be needed to confirm these associations and to better understand the underlying mechanisms.

Table 2 also shows the results of MLR analysis conducted to determine the potential influence of various factors on gene expression in TC patients where the meta-analysis analysis suggested the use of a random effects model. The possible confounding factors examined were country of origin and sample size, as well as the age of data publication used as a proxy for the study date. For each gene, the table lists the P-values for each of the three factors. The lower the P-value, the stronger the evidence for a significant association between the factor and the gene expression.

According to the results, the country of origin was significantly associated with CLDN1, TIMP1, and TIFF3 expression (P = 1.13E-05, P = 4.50E-03, and P = 5.51E-03, respectively). In contrast, LGALS1 was not significantly associated with any of the factors examined. The sample size was not significantly associated with any of the genes examined. The age of data collection was similarly not significantly associated with any of the genes examined. Since sample size and the age of data publication were not significantly associated with the expression of any of the other genes examined, only the country of origin would appear to be a major factor for gene expression variation in the patients of TC. This may very well reflect real variance among different population groups.

## Role of significant genes in meta-analysis in thyroid cancer

In order to investigate the links between the genes that exhibit significant expression changes in the meta-analysis (Table 2) and the two diseases, TC and SS, we constructed a shortest-path literature-based analysis between each gene and TC, as depicted in Fig 4.

The results indicated that six out of seven genes were previously implicated as biomarkers for TC consistent with their significance over many different studies in the meta-analysis presented here. Based on literature-data mining, the pathway demonstrated that all seven genes are involved, at least indirectly in promoting TC, and that SS activates four of them (PLA2G7, TFF3, LCN2, and CLDN1), while inhibiting three of them (LTF, TIMP1, and LGALS1). See S3 Table for details of the supporting references, which follow the same format as S1 Table. It is worth noting that the seven genes identified through expression meta-analysis do not overlap with those identified through literature data mining (Fig 1). However, five of the intermediate pathway genes (CCL2, CD274, LGALS3, BCL2, and TGFB1) were also among those identified in the previous literature-based analysis (Fig 1).

It is worth noting that the gene expression analysis of these seven genes using TCGA/GTEx thyroid carcinoma (THCA) data confirms the identification results from the meta-analysis of the GEO dataset, as shown in Fig 5A. Although PLA2G7 and LGALS1 did not reach statistical significance, all these genes exhibited the same direction of variation (increase or decrease) in both the GEO dataset and the THCA dataset.

Survival analysis revealed that only CLDN1 is significantly negatively associated with the risk of thyroid carcinoma (p(HR) = 0.031) (Fig 5B). Additionally, SS has been implicated in promoting CLDN1 (Fig 4), suggesting that SS may play a beneficial role in TC survival through the SS-CLDN1-TC pathway.

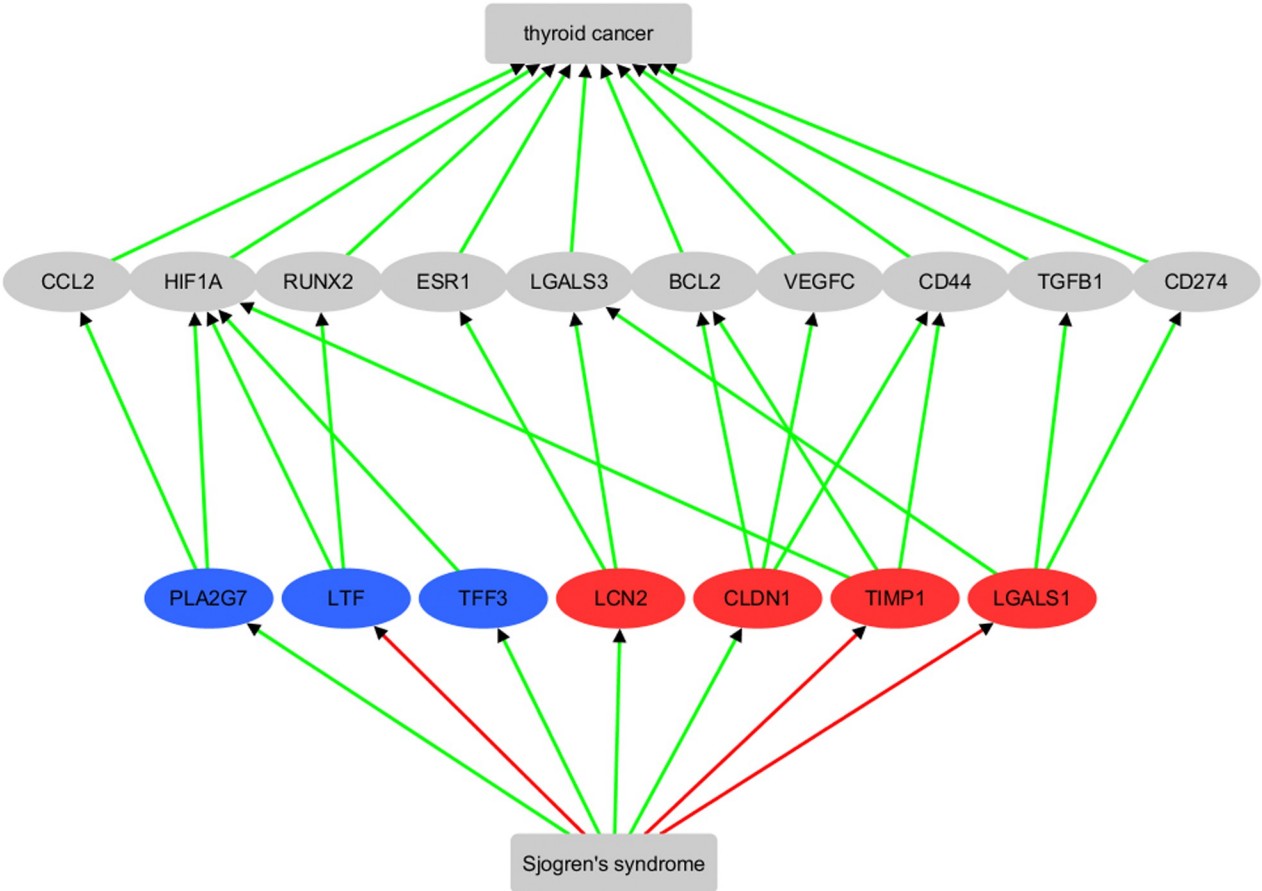

**Fig 4. Genes show significant expression change in meta-analysis and their connection with Sjogren's syndrome and thyroid cancer.** Edges in red represent a negative relationship, while those in green represent a positive relationship. A gene in red indicates significantly increased expression in the meta-analysis, whereas a gene in blue indicates significantly decreased expression in the meta-analysis.

### Enrichment analysis results

Enrichment analysis was performed on the combined genes from both the literature-based and the metanalysis representing a total of 21 molecules that are influenced by SS and affect TC in the two pathways presented in Figs 1 and 4. These were found to be significantly enriched in 78 pathways/gene sets (FDR corrected p-value < 0.0048; see **Supplementary Data**). Among these pathways, seven pathways/gene sets were related to cell growth/proliferation (with P-values ranging from 3.4e-05 to 0.002), two pathways/gene sets were related to the immune system (with P-values ranging from 0.0017 to 0.0027), and two pathways/gene sets were related to protein kinase (with P-values ranging from 0.0025 to 0.0031). Additionally, 12 out of the 21 genes were enriched in the top 10 GO terms (p-value < 7.7e-05 as presented in Fig 6A. These 12 genes consist of FOXP3, HMGB1, TGFB1, LGALS1, CD274, CCL2, PDCD1, LGALS3, BCL2, EP300, NTRK1, and MIR145, as shown in Fig 6B.

### Discussion

Despite multiple studies indicating a higher risk of TC in individuals with primary SS [9–12], the precise nature of the relationship between SS and TC is not yet fully understood and requires further investigation. In this study, we utilized extensive literature data mining and RNA

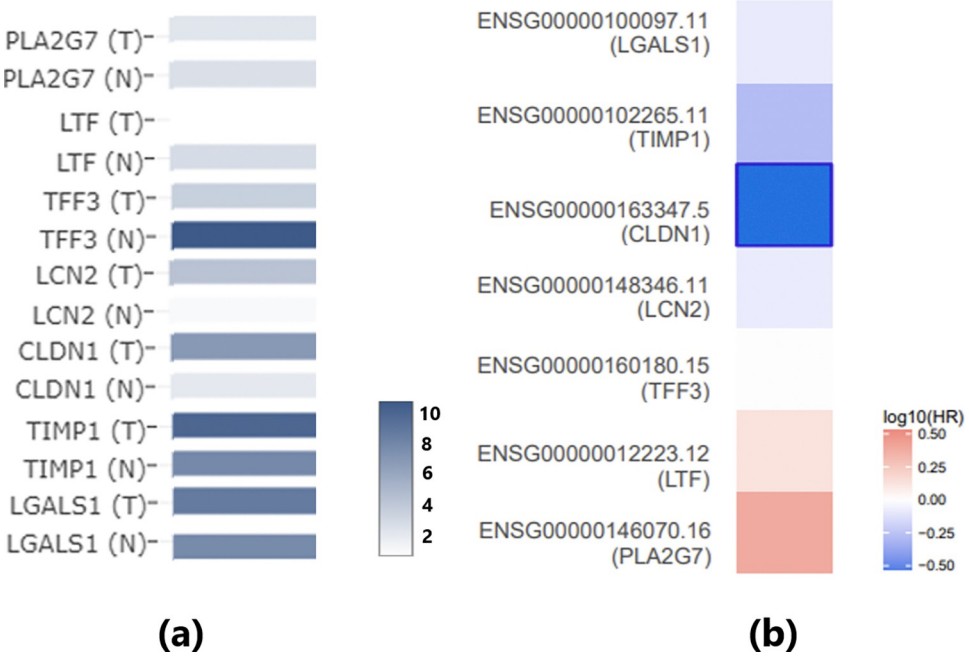

**Fig 5. Expression and survival analysis of meta-analysis-significant genes in the THCA dataset.** (a) Gene expression of the involved genes using the THCA dataset; (b) Survival map of these genes in thyroid carcinoma. Red indicates that higher expression levels are associated with increased risk (unfavorable prognosis), while blue indicates that higher expression levels are associated with reduced risk (favorable prognosis).

expression-based meta-analysis to investigate genes/proteins influenced by SS that may also contribute to the pathological development and progression of TC. Our analyses resulted in the construction of two molecular networks indicating a possible relationship between SS and TC.

The first molecular network constructed through large-scale literature data mining involved nine genes/proteins that contribute to the upregulation of TC. These genes/proteins are PDCD1, NTRK1, LGALS3, CD274, TGFB1, CCL2, CYP1A1, HMGB1, and EGF. Eight of these 9 genes are upregulated by SS making them clear candidates for mediating a risk effect of SS for the activation of TC. Studies have shown, for example, that PDCD1 expression is upregulated in the salivary glands of patients with SS [16], and PD-L1 (CD274) and PDCD1 expression are enhanced in salivary lymphocytes and salivary gland ductal and acinar epithelial cells of patients with SS [17]. The short-lived response and drug resistance of BRAF and MEK inhibitors for TC treatment can be improved by combining these inhibitors with PD-L1/PDCD1 antibodies [18]. In a 2019 study [19], the authors suggested that PD-L1 blockade might be a promising therapeutic strategy for TC management. They also mentioned the identification of PD-L1 in anaplastic TC, which could have direct therapeutic relevance to patients with refractory TC. Activation of PDCD1 and CD274 by SS. therefore, may quite likely affect the development and treatment of co-occurring TC. The other SS-driven TC-promoters may be acting through similar mechanisms.

SS may have an unfavorable impact on the management and progression of TC as well. Notably, two studies [20, 21] revealed a positive quantitative change in BCL2 expression associated with SS, as freshly isolated T cells from SS patients exhibited increased expression of BCL2. Given that BCL2 is known to safeguard thyroid carcinomas against apoptosis induced by chemotherapy [22], the SS-BCL2-TC pathway could represent a detrimental effect of SS on the treatment of TC.

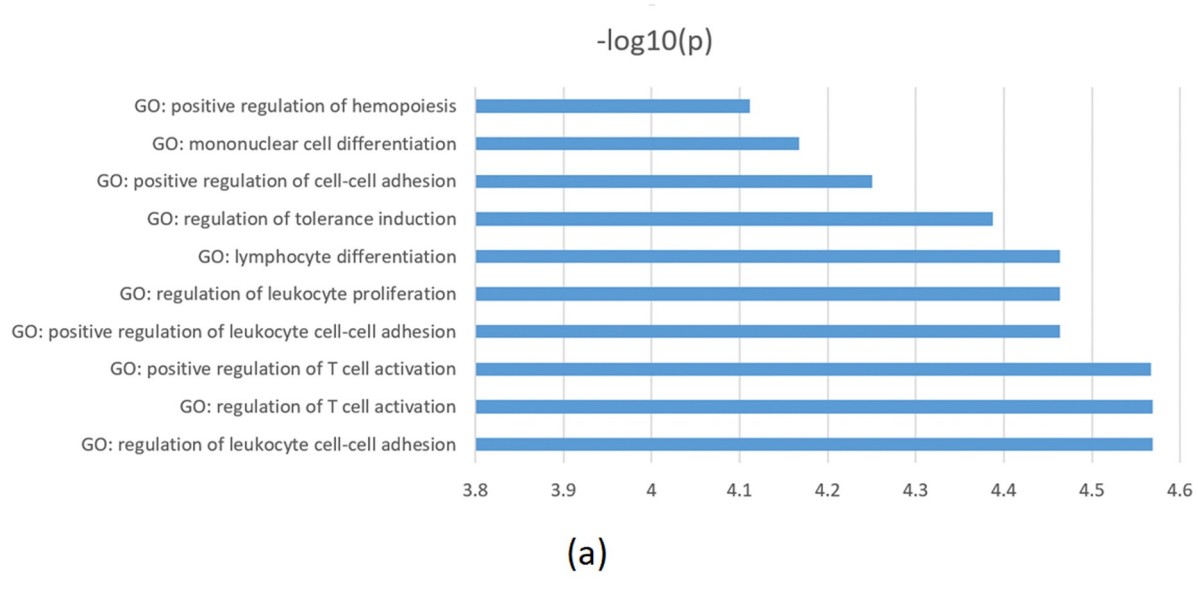

(a)

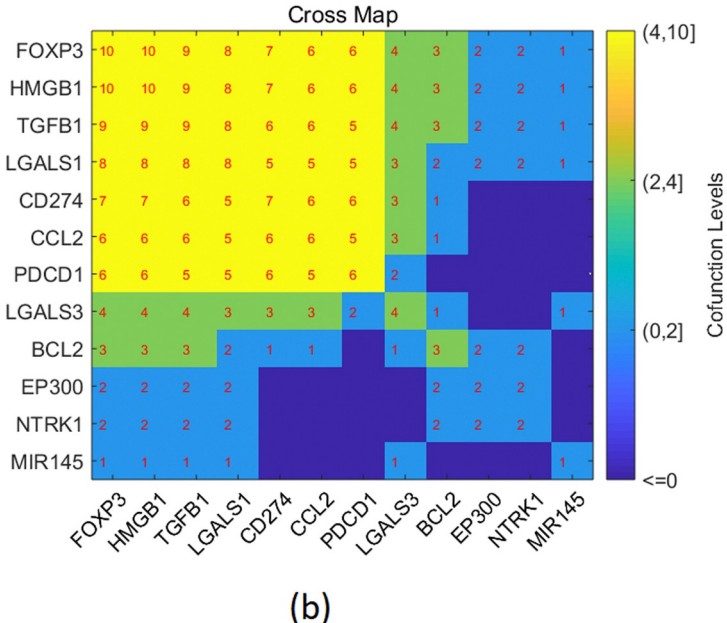

(b)

**Fig 6. Enrichment results using the 21 SS-driven genes influencing the pathology of TC.** (a) The top 10 GO terms enriched; (b) a co-occurrence index map of the genes enriched within the top 10 pathways; the numbers indicate the number of shared pathways between any two genes.

However, we note that SS may also play a protective role against TC through the inhibition of epidermal growth factor (EGF), a growth factor that plays an important role in the growth and differentiation of papillary carcinoma of the thyroid gland [23]. Several studies have shown that EGF enhances the migration and invasiveness of TC cells [24, 25], and its overexpression is associated with the development and progression of TC [26]. It has been suggested that SS may lead to a decrease in EGF levels [27, 28], resulting in a protective effect on the pathological development and progression of TC. The role of SS-inhibition of EGF remains unclear in the context of TC activation and progression.

The molecular pathway illustrated in Fig 1 also encompasses three TC inhibitors: MIR145, EP300, and MIR30C1. SS triggers the inhibition of all three of these genes/proteins. The inhibition of these natural regulators of TC may actually encourage its development and help to elevate the likelihood of TC progression. For instance, miR-145 has been suggested by several studies to act as a tumor suppressor in TC by inhibiting various pathways, including RAB5C, PI3K/AKT3, and protein kinase B [29–31]. A recent study [32] found that levels of miR-145 were decreased in salivary glands from SS patients, and this decrease was inversely correlated with the type I interferon score, mRNA levels of interferon-β, MUC1, toll-like receptor 4, and clinical parameters of SS patients (Ro/La autoantibodies and focus score). This suggests that the suppression of miR-145, a tumor suppressor, in SS could potentially worsen the prognosis for thyroid cancer.

Based on the meta-analysis, we identified seven TC regulated genes that show significant expression change in TC patients, suggesting that they may play a role in the pathological development of TC across multiple studies and different individual clinical scenarios. These genes included PLA2G7, LTF, TFF3, LCN2, CLDN1, TIMP1, and LGALS1. Literature-based pathway analysis showed that they are direct or indirect active regulators of TC, consistent with the meta-analysis results. Moreover, all of these genes/proteins can be shown (by literature analysis) to interact with positive promoters of TC, including CCL2, HIF1A, RUNX2, ESR1, LGALS3, BCL2, VEGFC, CD44, TGFB1, and CD274, as shown in Fig 4. By integrating the pathways presented in Figs 1 and 4, we identified three potential mechanisms through which SS may contribute to the pathological development of TC. These pathways include the SS-PLA2G7-CCL2-TC pathway, the SS-LCN2- LGALS3-TC pathway and the SS-CLDN1-BCL2-TC pathway.

As an autoimmune disorder, SS has been associated with an increased risk of TC, with particular emphasis on the regulation of CCL2 and PLA2G7. CCL2, a chemokine involved in inflammation, and PLA2G7, an enzyme linked to lipid metabolism, may interact to create a pro-inflammatory and pro-tumorigenic microenvironment conducive to TC progression. CCL2 is a key player in both SS and TC, where its association with SS disease activity and inflammation may occur through NF-κB pathway activation [33–35]. In TC, CCL2 promotes cell migration and is predictive of papillary thyroid carcinoma, highlighting its significance in tumor progression [36, 37]. Moreover, the modulation of CCL2 by factors such as Canagliflozin, PFHxA, USP15, and Vitamin D further connects its involvement in TC development, particularly in association with papillary thyroid carcinoma and Hashimoto's thyroiditis [36, 38, 39]. PLA2G7, influenced by SS-related factors like second-hand smoke, contributes to SS-associated inflammation and potentially interacts with CCL2 in COPD [40]. The expression and activity of PLA2G7 are also linked to B-cell lymphoma in SS, suggesting its diagnostic utility in SS-related malignancies [41]. The interplay between CCL2 and PLA2G7 in the context of SS and TC underscores the complex regulatory mechanisms underlying the pathogenesis and progression of these interconnected conditions, necessitating further research to elucidate their precise roles.

Moreover, the linkage between SS and an increased risk of TC may also be attributed to the regulation of LGALS3 (Galectin-3) and LCN2. LGALS3, known for promoting tumor growth and metastasis in TC, has shown significant associations with TC susceptibility, particularly in papillary thyroid carcinoma. Specific genetic variants of LGALS3 influence the risk of developing TC, and its role in the invasiveness of BRAF-mutated papillary thyroid carcinoma highlights its potential as a therapeutic target in tumor progression [42, 43]. LCN2, which regulates inflammation and immune response in the tumor microenvironment, is overexpressed in aggressive TC variants and is associated with epithelial-mesenchymal transition, making it a valuable predictive and diagnostic biomarker [44, 45]. The influence of both LGALS3 and

LCN2 on TC invasiveness via NF-κB pathways further underscores their therapeutic potential in specific TC subtypes [46, 47]. Additionally, the elevation of LGALS3 in SS stages suggests its role as an alarmin, particularly in SS patients with interstitial lung disease, and underscores the intricate interplay between LGALS3, LCN2, SS, and TC. This association between SS and an increased risk of TC, supported by higher incidence rates of TC in patients with primary SS, emphasizes the need for further research to elucidate the underlying mechanisms [48, 49].

In addition, SS upregulates CLDN1 and BCL2 [50, 51], two important genes involved in cell proliferation and apoptosis, which may contribute to an increased risk of TC. CLDN1, a tight junction protein, could disrupt normal cell-cell interactions in the thyroid gland, potentially promoting tumorigenesis [52, 53]. Meanwhile, BCL2, an anti-apoptotic protein, might prevent the programmed cell death of cancerous thyroid cells, contributing to tumorigenesis and therapeutic resistance [54]. Thus, the dysregulation of CLDN1 and BCL2 in the context of SS may contribute to the development and progression of TC by influencing cellular processes related to tumorigenesis and therapeutic resistance.

Enrichment analysis results (Fig 6A) of the combined genes from both the literature-based and the meta-analysis data indicated that these genes/proteins were primarily associated with the positive regulation of leukocytes and Treg cells [55–59]. While the elevated leukocyte levels can aid in eliminating TC cells, the increased levels of Treg cells may promote tolerance induction, which in turn can lead to the growth and evasion of the immune system by the tumor [60].

Expression analysis using TCGA/GTEx data confirms the expression changes of SS-driven genes that influence TC. Specifically, among the 14 genes in the first pathway (Fig 1), no significant changes were observed in TC patients compared to normal controls. However, seven genes showed consistent directional variation (either increased or decreased expression) across both the GEO dataset and the THCA dataset. These findings further validate the robustness of our meta-analysis.

Furthermore, survival analysis revealed that SS may beneficially influence TC survival through two distinct pathways. First, SS promotes CLDN1, which is significantly negatively associated with the risk of thyroid carcinoma (p(HR) = 0.031), suggesting a potential beneficial role in TC survival through the SS-CLDN1-TC pathway. Second, SS inhibits EGF, a promoter of TC that is significantly positively associated with the risk of thyroid carcinoma (p(HR) = 0.03). This aligns with the SS-EGF-TC pathway, further supporting the notion that SS may have a protective role in TC survival. These findings highlight the complex and multifaceted relationship between SS and TC, warranting further investigation.

In summary, this study has notable strengths and limitations that should be addressed in future research. A key strength is the integration of literature data mining with GEO database analysis, which allowed us to explore the potential genetic and molecular contributions of SS to TC. This approach gathered information from a wide range of studies, potentially uncovering hidden evidence of SS-TC interactions. We identified multiple pathways and highlighted three possible mechanisms—the SS-PLA2G7-CCL2-TC pathway, the SS-LCN2-LGALS3-TC pathway, and the SS-CLDN1-BCL2-TC pathway—that may explain how SS contributes to TC development. However, the study's limitations include its reliance on literature and existing TC expression databases, which lack specific SS disease context. Experimental data from SS patients who develop TC should be collected for further validation of the identified genes and pathways. Furthermore, in addition to the individual evaluation of significant genes, a gene signature analysis considering the group of target genes should be conducted to assess their collective significance in TC and SS, particularly in relation to disease progression, severity, and treatment response.

## Conclusion

Although LDM-based pathway analysis reveals potential pathways through which SS may adversely influence TC progression and development—such as the SS-PLA2G7-CCL2-TC pathway, the SS-LCN2-LGALS3-TC pathway, and the SS-CLDN1-BCL2-TC pathway—SS may also have a beneficial impact on TC survival through the CLDN1 and EGF pathways. These findings underscore the complex relationship between SS and TC, warranting further investigation.

## Supporting information

**S1 Table. References supporting the relationships presented in Fig 1.**
(XLSX)

**S2 Table. Meta-analysis results for 245 genes using 16 TC expression datasets.**
(XLSX)

**S3 Table. References supporting the influnce of SS on seven genes that show significance in Meta-analysis using 16 TC expression datasets.**
(XLSX)

## Author Contributions

**Data curation:** Zhen Wu, Jiaming Chen.

**Formal analysis:** Xixi Shen.

**Methodology:** Qian Shi, Lizhen Hou.

**Supervision:** Meng Lian.

**Writing – original draft:** Fanyong Kong, Boxuan Han.

**Writing – review & editing:** Jugao Fang.

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
