## [Decision Letter · Decision Letter 0]

15 Jul 2024

PONE-D-24-14027Explore the possible influence of Sjogren's syndrome on Thyroid CancerPLOS ONE

Dear Dr. lian,

Thank you for submitting your manuscript to PLOS ONE. After careful consideration, we feel that it has merit but does not fully meet PLOS ONE’s publication criteria as it currently stands. Therefore, we invite you to submit a revised version of the manuscript that addresses the points raised during the review process.

We look forward to receiving your revised manuscript.

Kind regards,

Vahid Mansouri, M.D.

Academic Editor

PLOS ONE

Journal Requirements:

“This work was partially supported by the Beijing Municipal Administration of Hospitals’ Youth Programme (QMS20210206); Scientific Research Common Program of Beijing Municipal Commission of Education (KM202210025014).”

Additional Editor Comments:

Hello

Thank you for submitting your manuscript in PLOS one.

After trying multiple times, invited reviewers did not accept to review the submitted manuscript. So, this time I am sending comments of only one reviewer.

Please be aware that we continue to invite more reviewers to have their comments on your manuscript in next round of peer-review. Also, please let us know, about any suggested suitable reviewers in field.

Please, consider the reviewers comments and respond to them one by one, including any necessary changes.

Reviewers' comments:

Reviewer's Responses to Questions

**Comments to the Author**

1. Is the manuscript technically sound, and do the data support the conclusions?

Reviewer #1: Yes

2. Has the statistical analysis been performed appropriately and rigorously? 

Reviewer #1: Yes

3. Have the authors made all data underlying the findings in their manuscript fully available?

Reviewer #1: No

4. Is the manuscript presented in an intelligible fashion and written in standard English?

Reviewer #1: Yes

5. Review Comments to the Author

Reviewer #1: This study mainly used literature data mining and GEO database joint analysis to explore the common genes/proteins between SS and TC, which has a certain degree of innovation. However, the methods of literature data mining in the study were not specifically specified, which raises doubts about the subsequent results. And this study only used relevant information from literature and databases, without further experimental validation analysis of the excavated genes or validation analysis of other databases. The reliability and authenticity of the final results are questionable. I think it is necessary to conduct further validation analysis, whether it is laboratory related analysis such as PCR, WB, IHC, or selecting other databases for validation, such as TC related TGCA database or SS related GEO database.

Here are some specific questions about the article:

1. In methods and materials,, the relevant driving factors extracted from literature data mining, and the specific methods of literature data mining? What literature databases have been mined? What are these documents specifically? It is not explicitly stated in the article. These steps are necessary as they relate to the accuracy of subsequent analysis results and the rigor and rationality of the article.

2. How can we determine if PDCD1, NTRK1, LGALS3, CD274, FOXP3, BCL2, CYP1130 HMGB1, TGFB1, CCL2, as well as MIR145, EP300, and FOXP3 are activators/inhibitors of TC? What are the relevant literature? Specific instructions are required.

3. How many screened LEF genes are in each of the 16 TC expression databases? How did they end up getting at these genes in Table 2? Need to list them specifically.

4. What is the basis for the statement in line 200 that based on literature data mining, SS activated four of these genes (PLA2G7, TFF3, LCN2 and CLDN1) while inhibiting three of them (LTF, TIMP1 and LGALS1)? Relevant literature is needed to support this result.

5. The serial numbers in the article are confusing, e.g., 1.result appears in line 125, but 3.1 driven molecules pathway influencing TC appears in line 126. The format of the article needs to be systematically checked.

6. The discussion is merely a description of the results and a stack of literature citations, and should focus on describing the key genes in which SS is associated with TC. For example, it is mentioned in the text that CCL2 is up-regulated by SS and PLA2G7 is significantly changed in TC. PLA2G7 can be positively correlated with TC through CCL2 and HIF1A, and CCL2 has been proved to promote metastasis of thyroid cancer. It would be a better direction to summarize the SS-CCL2-PLA2G7-TC pathway and conduct an in-depth discussion and experimental or database validation of this pathway.

7. The description of the conclusion is too broad, and the writing structure is incorrect. Specific research points in the article should be summarized, such as what is the relationship between SS and TC? Which common gene affects the incidence of both? As well as the strengths and weaknesses of the article should be specifically described in the conclusion.

6. PLOS authors have the option to publish the peer review history of their article (what does this mean?). If published, this will include your full peer review and any attached files.

Reviewer #1: **Yes: **Xinchang Wang

---

## [Author Response · Author response to Decision Letter 0]

17 Sep 2024

Journal Requirements:

and

Answer: We check the format of the revised manuscript and made necessary changes according to the PLOS ONE style templates. Please let us know if further changes are needed.

“This work was partially supported by the Beijing Municipal Administration of Hospitals’ Youth Programme (QMS20210206); Scientific Research Common Program of Beijing Municipal Commission of Education (KM202210025014).”

Answer: We have the financial statement revised as follows.

This work was partially supported by the Beijing Municipal Administration of Hospitals’ Youth Programme (QMS20210206); Scientific Research Common Program of Beijing Municipal Commission of Education (KM202210025014). The funders had no role in study design, data collection and analysis, decision to publish, or preparation of the manuscript.

Reviewer #1: This study mainly used literature data mining and GEO database joint analysis to explore the common genes/proteins between SS and TC, which has a certain degree of innovation. However, the methods of literature data mining in the study were not specifically specified, which raises doubts about the subsequent results. And this study only used relevant information from literature and databases, without further experimental validation analysis of the excavated genes or validation analysis of other databases. The reliability and authenticity of the final results are questionable. I think it is necessary to conduct further validation analysis, whether it is laboratory related analysis such as PCR, WB, IHC, or selecting other databases for validation, such as TC related TGCA database or SS related GEO database.

Here are some specific questions about the article:

1. In methods and materials, the relevant driving factors extracted from literature data mining, and the specific methods of literature data mining? What literature databases have been mined? What are these documents specifically? It is not explicitly stated in the article. These steps are necessary as they relate to the accuracy of subsequent analysis results and the rigor and rationality of the article.

Answer: Thank you for your suggestion. We fully agree that providing this specific information is essential for the audience to understand the accuracy of the subsequent analysis results, as well as the rigor and rationality of the article. We have added more details in the Methods section under ‘SS-driven molecules influencing TC’ and cited the following text:

To investigate the potential influence of SS on TC, we performed literature data mining (LDM) to identify SS-driven molecules that also act as upstream regulators of TC. The LDM was carried out using Pathway Studio (version 12.3), which utilizes the natural language processing (NLP) tool MedScan [13]. This process involved mining data from 24 million PubMed abstracts and 3.5 million Elsevier and third-party full-text papers. Each relationship or edge was established based on facts extracted from the literature using NLP technology, supported by at least one reference. Key information extracted included relation type, relation polarity, reference title, PMID/DOI, and the key sentences where the relationship was identified.

2. How can we determine if PDCD1, NTRK1, LGALS3, CD274, FOXP3, BCL2, CYP1130 HMGB1, TGFB1, CCL2, as well as MIR145, EP300, and FOXP3 are activators/inhibitors of TC? What are the relevant literature? Specific instructions are required.

Answer: Thank you for the question. The identification of these molecules as activators or inhibitors of TC was determined by a Natural Language Processing (NLP) algorithm. Additionally, the relevant information, including specific sentences from the literature, was extracted for manual validation. To enhance transparency and allow for validation by the audience, we included a Reference table in Supplementary Material S1 Table alongside Figure 1. This table provides detailed information such as the Relation Name, Reference, Title, related Sentence, and supporting references (e.g., Source, PMID, PubYear, and DOI). We have also added a detailed description in the Results section under ‘SS-driven molecules pathway influencing TC’ and cited the following text:

For details on the relationships/edges presented in Fig 1, please refer to Supplementary Material S1 Table, which includes information on the supporting references, such as the reference title, PMID, DOI, and the relevant sentences where the relationships were identified.

3. How many screened LEF genes are in each of the 16 TC expression databases? How did they end up getting at these genes in Table 2? Need to list them specifically.

Answer: Thank you for your question. We screened 273 SS-driven molecules and analyzed their expression changes in TC patients compared to controls using 16 TC expression datasets. Out of these 273 molecules, 245 were identified in at least one of the 16 datasets. Except for CD68 and NEAT1, all other genes were identified in at least 10 of the TC expression datasets. The genes listed in Table 2 were selected through a meta-analysis of these 245 genes. We calculated the log2 fold-change (LFC) and p-value across all datasets to determine significance, with genes selected based on an absolute LFC greater than or equal to 1 and a p-value less than 0.05. Depending on the heterogeneity of the datasets, either a random-effects or fixed-effect model was used.

The detailed results for each of these genes, including the meta-analysis model used, the number of datasets (studies) included, and the results of multiple-linear regression analysis, are provided in Supplementary Material S2 Table. This supplementary material ensures transparency and allows for verification of the screening process.

We added a description in the results section "Meta-analysis results," as follows:

"For the 273 SS-driven molecules, 245 were identified in at least one of the 16 TC expression datasets. Except for CD68 and NEAT1, all other genes were identified in at least 10 TC expression datasets. Detailed meta-analysis results for each of these genes are provided in Supplementary Material S2 Table, including the meta-analysis model used, the number of datasets (studies) involved, LFC, p-value, and the results of multiple-linear regression analysis."

4. What is the basis for the statement in line 200 that based on literature data mining, SS activated four of these genes (PLA2G7, TFF3, LCN2 and CLDN1) while inhibiting three of them (LTF, TIMP1 and LGALS1)? Relevant literature is needed to support this result.

Answer: Thank you for the question. We have provided the relevant literature to support this statement in Supplementary Material S3 Table. This material includes detailed references that substantiate the activation of PLA2G7, TFF3, LCN2, and CLDN1 by SS, as well as the inhibition of LTF, TIMP1, and LGALS1. We have added a description in the relevant Results section, "Role of significant genes in meta-analysis in thyroid cancer," and cited it as follows:

"… and that SS activates four of them (PLA2G7, TFF3, LCN2, and CLDN1), while inhibiting three of them (LTF, TIMP1, and LGALS1). See Supplementary Material S3 Table for the details of the supporting references."

5. The serial numbers in the article are confusing, e.g., 1.result appears in line 125, but 3.1 driven molecules pathway influencing TC appears in line 126. The format of the article needs to be systematically checked.

Answer: Thank you for pointing this out. We have systematically reviewed and reformatted the article to ensure consistency and adherence to PLOS ONE's style requirements. All serial numbers and section headings have been corrected to improve clarity and organization.

6. The discussion is merely a description of the results and a stack of literature citations, and should focus on describing the key genes in which SS is associated with TC. For example, it is mentioned in the text that CCL2 is up-regulated by SS and PLA2G7 is significantly changed in TC. PLA2G7 can be positively correlated with TC through CCL2 and HIF1A, and CCL2 has been proved to promote metastasis of thyroid cancer. It would be a better direction to summarize the SS-CCL2-PLA2G7-TC pathway and conduct an in-depth discussion and experimental or database validation of this pathway.

Answer: Thank you for the valuable suggestion. Based on the Literature Data Mining (LDM) and Meta-analysis using TC expression datasets, we identified potential pathways that may explain how SS contributes to the pathological development of TC, as illustrated in Figures 1 and 3. By integrating these pathways, we highlighted three key mechanisms—the SS-PLA2G7-CCL2-TC pathway, the SS-LCN2- LGALS3-TC pathway and the SS-CLDN1-BCL2-TC pathway—and conducted an in-depth discussion on their possible mechanisms. We have incorporated these discussions into the Discussion section, as cited below: 

By integrating the pathways presented in Figures 1 and 3, we identified three potential mechanisms through which SS may contribute to the pathological development of TC. These pathways include the SS-PLA2G7-CCL2-TC pathway, the SS-LCN2- LGALS3-TC pathway and the SS-CLDN1-BCL2-TC pathway.

As an autoimmune disorder, SS has been associated with an increased risk of TC, with particular emphasis on the regulation of CCL2 and PLA2G7. CCL2, a chemokine involved in inflammation, and PLA2G7, an enzyme linked to lipid metabolism, may interact to create a pro-inflammatory and pro-tumorigenic microenvironment conducive to TC progression. CCL2 is a key player in both SS and TC, where its association with SS disease activity and inflammation may occur through NF-κB pathway activation [32-34]. In TC, CCL2 promotes cell migration and is predictive of papillary thyroid carcinoma, highlighting its significance in tumor progression [35, 36]. Moreover, the modulation of CCL2 by factors such as Canagliflozin, PFHxA, USP15, and Vitamin D further connects its involvement in TC development, particularly in association with papillary thyroid carcinoma and Hashimoto's thyroiditis [35, 37, 38]. PLA2G7, influenced by SS-related factors like second-hand smoke, contributes to SS-associated inflammation and potentially interacts with CCL2 in COPD [39]. The expression and activity of PLA2G7 are also linked to B-cell lymphoma in SS, suggesting its diagnostic utility in SS-related malignancies [40]. The interplay between CCL2 and PLA2G7 in the context of SS and TC underscores the complex regulatory mechanisms underlying the pathogenesis and progression of these interconnected conditions, necessitating further research to elucidate their precise roles.

Moreover, the linkage between SS and an increased risk of TC may also be attributed to the regulation of LGALS3 (Galectin-3) and LCN2. LGALS3, known for promoting tumor growth and metastasis in TC, has shown significant associations with TC susceptibility, particularly in papillary thyroid carcinoma. Specific genetic variants of LGALS3 influence the risk of developing TC, and its role in the invasiveness of BRAF-mutated papillary thyroid carcinoma highlights its potential as a therapeutic target in tumor progression [41, 42]. LCN2, which regulates inflammation and immune response in the tumor microenvironment, is overexpressed in aggressive TC variants and is associated with epithelial-mesenchymal transition, making it a valuable predictive and diagnostic biomarker [43, 44]. The influence of both LGALS3 and LCN2 on TC invasiveness via NF-κB pathways further underscores their therapeutic potential in specific TC subtypes [45, 46]. Additionally, the elevation of LGALS3 in SS stages suggests its role as an alarmin, particularly in SS patients with interstitial lung disease, and underscores the intricate interplay between LGALS3, LCN2, SS, and TC. This association between SS and an increased risk of TC, supported by higher incidence rates of TC in patients with primary SS, emphasizes the need for further research to elucidate the underlying mechanisms [47, 48].

In addition, SS upregulates CLDN1 and BCL2 [49, 50], two important genes involved in cell proliferation and apoptosis, which may contribute to an increased risk of TC. CLDN1, a tight junction protein, could disrupt normal cell-cell interactions in the thyroid gland, potentially promoting tumorigenesis [51, 52]. Meanwhile, BCL2, an anti-apoptotic protein, might prevent the programmed cell death of cancerous thyroid cells, contributing to tumorigenesis and therapeutic resistance [53]. Thus, the dysregulation of CLDN1 and BCL2 in the context of SS may contribute to the development and progression of TC by influencing cellular processes related to tumorigenesis and therapeutic resistance.

7. The description of the conclusion is too broad, and the writing structure is incorrect. Specific research points in the article should be summarized, such as what is the relationship between SS and TC? Which common gene affects the incidence of both? As well as the strengths and weaknesses of the article should be specifically described in the conclusion.

Answer: Thank you for the suggestion. We have revised the Conclusion to incorporate the major findings of the study. The revised Conclusion is now cited as follows.

Conclusion

Our study proposed multiple pathways and highlighted three possible mechanisms—the SS-PLA2G7-CCL2-TC pathway, the SS-LCN2-LGALS3-TC pathway, and the SS-CLDN1-BCL2-TC pathway—that may explain how SS contributes to the pathological development of TC. These findings provide valuable insights into the relationship between primary SS and TC and underscore the need for further investigation to fully understand the precise nature of this relationship and its potential implications for TC diagnosis and treatment.

We also included a discussion of the strengths and limitations of this study, which are cited as follows.

In summary, this study has notable strengths and limitations that should be addressed in future research. A key strength is the integration of literature data mining with GEO database analysis, which allowed us to explore the potential genetic and molecular contributions of SS to TC. This approach gathered information from a wide range of studies, potentially uncovering hidden evidence of SS-TC interactions. We identified multiple pathways and highlighted three possible mechanisms—the SS-PLA2G7-CCL2-TC pathway, the SS-LCN2-LGALS3-TC pathway, and the SS-CLDN1-BCL2-TC pathway—that may explain how SS contributes to TC development. However, the study's limitations include its reliance on literature and existing TC expression databases, which lack specific SS disease context. Experimental data from SS patients who develop TC should be collected for further validation of the identified genes and pathways.

Moreover, based on these changes, we have revised the Abstract, which is now cited as follows.

Abstract

Objectives

To explore the potential influence of Sjogren's syndrome (SS) on thyroid cancer (TC).

Methods

First, literature data mining was conducted to reconstruct functional pathways connecting SS and TC. Subsequently, a meta-analysis was performed using 16 TC case/control expression datasets to investigate the expression changes of genes mediated by SS. Finally, gene set enrichment analysis (GSEA) was conducted on the significant genes identified in the expression meta-analysis to identify potential pathways and functi

---

## [Decision Letter · Decision Letter 1]

13 Nov 2024

PONE-D-24-14027R1Explore the possible influence of Sjogren's syndrome on Thyroid CancerPLOS ONE

Dear Dr. lian,

Thank you for submitting your manuscript to PLOS ONE. After careful consideration, we feel that it has merit but does not fully meet PLOS ONE’s publication criteria as it currently stands. Therefore, we invite you to submit a revised version of the manuscript that addresses the points raised during the review process.

We look forward to receiving your revised manuscript.

Kind regards,

Vahid Mansouri, M.D.

Academic Editor

PLOS ONE

Reviewers' comments:

Reviewer's Responses to Questions

**Comments to the Author**

1. If the authors have adequately addressed your comments raised in a previous round of review and you feel that this manuscript is now acceptable for publication, you may indicate that here to bypass the “Comments to the Author” section, enter your conflict of interest statement in the “Confidential to Editor” section, and submit your "Accept" recommendation.

Reviewer #2: (No Response)

Reviewer #3: (No Response)

2. Is the manuscript technically sound, and do the data support the conclusions?

Reviewer #2: Partly

Reviewer #3: (No Response)

3. Has the statistical analysis been performed appropriately and rigorously? 

Reviewer #2: Yes

Reviewer #3: (No Response)

4. Have the authors made all data underlying the findings in their manuscript fully available?

Reviewer #2: Yes

Reviewer #3: (No Response)

5. Is the manuscript presented in an intelligible fashion and written in standard English?

Reviewer #2: Yes

Reviewer #3: (No Response)

6. Review Comments to the Author

Reviewer #2: This study presents an investigation into the potential molecular pathways linking Sjogren's syndrome (SS) and thyroid cancer (TC). The authors have integrated literature data mining and RNA expression-based meta-analysis to identify genes influenced by SS that may contribute to the pathological development and progression of TC.

However, the comments from Reviewer #1 on the database validation of the highlighted genes and pathways have not been sufficiently addressed. Since the goal of this study is to investigate genes influenced by SS that may also contribute to the pathological development and progression of TC [Lines 74-75, 243-244, 261, 277-278, etc.], the authors need to further investigate these genes using data from TCGA/PanCancer Atlas.

The TCGA has close to 500 TC samples with gene expression data. The GEPIA 2 portal [Tang et al., 2019; 10.1093/nar/gkz430] has readily-available differential expression analysis of TC samples integrated with healthy controls from GTEx. Furthermore, the genes highlighted in the study should be further investigated. For instance, are there hotspot mutations or high mutation-burden in these SS-related genes? Is the expression of these genes linked with severity and/or survival? This information is also readily accessible through cBioPortal and other portals. This will greatly improve the robustness of the findings in the current work by commenting on prognosis or other characteristics of these SS-linked genes in TC.

Additional comments are as follows:

Lines 82-85: Approximately when was the literature data mining performed (month, year)? This will indicate the status of the corpus from PubMed and Elsevier included in the mining.

Line 99: Why were expression datasets limited to arrays? Newer, high-throughput/NGS, technologies have higher sensitivities and should be included in the current study.

Line 100: Was the filtering criteria SS vs. healthy or TC vs. healthy?

Reviewer #3: Cleared by editor as sufficiently addressed.

7. PLOS authors have the option to publish the peer review history of their article (what does this mean?). If published, this will include your full peer review and any attached files.

Reviewer #2: No

Reviewer #3: No

---

## [Author Response · Author response to Decision Letter 1]

19 Dec 2024

We appreciate the comments and suggestions from the reviewer and Editor. We made changes accordingly and provided a point-by-point answer for each question in the file 'Response to Reviewers.docx'. Please let us know if further changes are needed.

---

## [Decision Letter · Decision Letter 2]

10 Jan 2025

PONE-D-24-14027R2Explore the possible influence of Sjogren's syndrome on Thyroid CancerPLOS ONE

Dear Dr. lian,

Thank you for submitting your manuscript to PLOS ONE. After careful consideration, we feel that it has merit but does not fully meet PLOS ONE’s publication criteria as it currently stands. Therefore, we invite you to submit a revised version of the manuscript that addresses the points raised during the review process.

We look forward to receiving your revised manuscript.

Kind regards,

Vahid Mansouri, M.D.

Academic Editor

PLOS ONE

**Journal Requirements:**

Reviewers' comments:

Reviewer's Responses to Questions

**Comments to the Author**

1. If the authors have adequately addressed your comments raised in a previous round of review and you feel that this manuscript is now acceptable for publication, you may indicate that here to bypass the “Comments to the Author” section, enter your conflict of interest statement in the “Confidential to Editor” section, and submit your "Accept" recommendation.

Reviewer #1: All comments have been addressed

Reviewer #2: All comments have been addressed

Reviewer #3: All comments have been addressed

2. Is the manuscript technically sound, and do the data support the conclusions?

Reviewer #1: Yes

Reviewer #2: (No Response)

Reviewer #3: Yes

3. Has the statistical analysis been performed appropriately and rigorously? 

Reviewer #1: Yes

Reviewer #2: (No Response)

Reviewer #3: Yes

4. Have the authors made all data underlying the findings in their manuscript fully available?

Reviewer #1: Yes

Reviewer #2: (No Response)

Reviewer #3: Yes

5. Is the manuscript presented in an intelligible fashion and written in standard English?

Reviewer #1: Yes

Reviewer #2: (No Response)

Reviewer #3: Yes

6. Review Comments to the Author

**Reviewer #1:** Thank you for the opportunity to review this manuscript. After carefully evaluating the revised version, I find that the authors have addressed all the review comments thoroughly and have made significant improvements to the quality and clarity of the manuscript.

I am satisfied with the revisions and recommend the manuscript for acceptance and publication.

**Reviewer #2:** Thank you for submitting a revised manuscript and for addressing the major comments. For future, a gene set/signature analysis may also be considered when assessing a group of genes (e.g., 14 genes from Figure 1a) with disease progression/severity/treatment response/etc., as seldom do individual genes show striking correlation with disease/clinical characteristics.

**Reviewer #3: **This paper has improved a lot now. I have some minor suggestions.

1) Authors should revise the title and embody the detailed work you have conducted, for example: Explore the possible influence of sjogren's syndrome on thyroid cancer: a literature data mining and meta-analysis.

2) Authors should rearrange and restructure all the figures to make them look more sightly.

7. PLOS authors have the option to publish the peer review history of their article (what does this mean?). If published, this will include your full peer review and any attached files.

Reviewer #1: No

Reviewer #2: No

Reviewer #3: No

---

## [Author Response · Author response to Decision Letter 2]

10 Jan 2025

We have made the necessary changes based on the reviewer comments and provided a detailed point-by-point response, which has been uploaded in the file titled Response to Reviewers.docx. Please let us know if any further revisions are required.

---

## [Decision Letter · Decision Letter 3]

16 Jan 2025

PONE-D-24-14027R3

Explore the Possible Influence of Sjogren's Syndrome on Thyroid Cancer: A Literature Data Mining and Meta-Analysis

PLOS ONE

Dear Dr. lian,

Thank you for submitting your manuscript to PLOS ONE. After careful consideration, we feel that it has merit but does not fully meet PLOS ONE’s publication criteria as it currently stands. Therefore, we invite you to submit a revised version of the manuscript that addresses the points raised during the review process.

We look forward to receiving your revised manuscript.

Kind regards,

Vahid Mansouri, M.D.

Academic Editor

PLOS ONE

Journal Requirements:

Additional Editor Comments:

Thank you for addressing all the reviewer comments.

The manuscript can be accepted after this revision. Just there are some minor points (that has been already mentioned) that needs further consideration:

- Please redraw the figure 1 and figure 4. The arrows in these figures is not clear. Consider using more solid arrows that are not overlapping. Also please be careful about the names and the shapes. These figures can be more stylish and informative. Please provide the figures according to the journal requirements.

- What is the reason for highlighting some items in the Table S1?

- Please double check the citation of Tables throughout the text. I recommend to give more description on the supplementary data.

- Please double check the affiliations and the spelling of the authors.

- Please also check all the references for being valid and accurate in the journal formatting.

Reviewers' comments:

Reviewer's Responses to Questions

**Comments to the Author**

1. If the authors have adequately addressed your comments raised in a previous round of review and you feel that this manuscript is now acceptable for publication, you may indicate that here to bypass the “Comments to the Author” section, enter your conflict of interest statement in the “Confidential to Editor” section, and submit your "Accept" recommendation.

Reviewer #3: All comments have been addressed

2. Is the manuscript technically sound, and do the data support the conclusions?

Reviewer #3: Yes

3. Has the statistical analysis been performed appropriately and rigorously? 

Reviewer #3: Yes

4. Have the authors made all data underlying the findings in their manuscript fully available?

Reviewer #3: Yes

5. Is the manuscript presented in an intelligible fashion and written in standard English?

Reviewer #3: Yes

6. Review Comments to the Author

Reviewer #3: I have no further comments now, and I think this paper can be accepted now. Thanks for authors' response to my concerns.

7. PLOS authors have the option to publish the peer review history of their article (what does this mean?). If published, this will include your full peer review and any attached files.

Reviewer #3: No

---

## [Author Response · Author response to Decision Letter 3]

19 Jan 2025

We have provided a point-by-point response to the Editor's comments in the file Response to Reviewers.docx. Please let us know if further changes are required.

---

## [Editor Report · Decision Letter 4]

21 Jan 2025

Explore the Possible Influence of Sjogren's Syndrome on Thyroid Cancer: A Literature Data Mining and Meta-Analysis

PONE-D-24-14027R4

Dear Dr. lian,

We’re pleased to inform you that your manuscript has been judged scientifically suitable for publication and will be formally accepted for publication once it meets all outstanding technical requirements.

Kind regards,

Vahid Mansouri, M.D.

Academic Editor

PLOS ONE
---

## [Editor Report · Acceptance letter]

30 Jan 2025

PONE-D-24-14027R4 

PLOS ONE

Dear Dr. Lian, 

I'm pleased to inform you that your manuscript has been deemed suitable for publication in PLOS ONE. Congratulations! Your manuscript is now being handed over to our production team.

Kind regards, 

on behalf of

Dr. Vahid Mansouri 

Academic Editor

PLOS ONE